# COVID-19 pandemic and food security in different contexts: A systematic review protocol

**Azam Doustmohammadian[1], Fatemeh Mohammadi-Nasrabadi ⬡[2]\*, Ghasem Fadavi[3]**

**1** Gastrointestinal and Liver Disease Research Center (GILDRC), Iran University of Medical Sciences, Tehran, Iran, **2** Food, Halal and Agricultural Products Research Group, Research Center of Food Technology and Agricultural Products, Standard Research Institute, Karaj, Alborz, Iran, **3** Food and Nutrition Policy and Planning Research Department, National Nutrition and Food Technology Research Institute (NNFTRI), Faculty of Nutrition Sciences and Food technology, Shahid Beheshti University of Medical Sciences, Tehran, Iran

\* f.mohammadinasrabadi@sbmu.ac.ir

**Editor:** Negar Rezaei, Non-Communicable Diseases Research Center, Endocrinology and Metabolism Population Sciences Institute, Tehran University of Medical Sciences, ISLAMIC REPUBLIC OF IRAN

## Abstract

### Background

Given the unprecedented nature of the COVID-19 crisis and the importance of early implementation of prevention programs, it is essential to understand better its potential impacts on various food security dimensions and indicators.

### Methods

Research databases, including Cochrane Central Register of Controlled Trials (CENTRAL), Cochrane Public Health Register, PubMed/MEDLINE, SCOPUS, EMBASE, CINAHL, Web of science, and Google Scholar, will be searched using a search strategy and keywords developed in collaboration with librarians. The review will include all field and community trials and observational studies in all population groups. Searching electronic databases, study selection, and data extraction will be conducted by two researchers independently. Four critical appraisal tools will be used to assess the quality of included studies according to the study design: The Joanna Briggs Institute (JBI) Prevalence Critical Appraisal tool, the JBI critical appraisal checklist for randomized control/pseudo-randomized trial, descriptive/ case series, and comparable cohort/case-control. These tools were initially designed for use in systematic reviews. A narrative synthesis will be implemented to summarize findings if meta-analyses are not appropriate.

### Discussion

The current systematic review results can be used to predict the effect of COVID-19 on the individuals' and households' food security, especially in vulnerable populations, and develop effective interventions. This review can provide information for policymakers to better understand the factors influencing the implementation of these interventions and inform decision-making to improve food security.

**Data Availability Statement:** The research data will be made publicly available when the study is completed and published.

**Funding:** This systematic review is funded by National Nutrition and Food Technology Research Institute, Shahid Beheshti University of Medical Sciences (Grant No. 99-23953). The funders had and will not have a role in study design, data collection and analysis, decision to publish, or preparation of the manuscript except for reviewing the proposal.

**Competing interests:** The authors have declared that no competing interests exist.

**Abbreviations:** COVID-19, Coronavirus Disease 2019; PRISMA, Preferred Reporting Items for Systematic Reviews and Meta-Analyses; PRISMA-P, Preferred Reporting Items for Systematic Review and Meta-Analysis Protocols; SRA-DM, Systematic Review Assistant-Deduplication Module; NOS, Newcastle-Ottawa quality assessment scale.

## Systematic review registration

The present study registration number in the international prospective register of systematic reviews (PROSPERO) is CRD42020185843.

## 1. Introduction

### 1.1. Description of the condition

The Coronavirus Disease 2019 (COVID-19) pandemic is a health crisis threatening the food and nutrition security of millions of people worldwide. COVID-19 affects the four dimensions of food and nutrition security: availability, accessibility, consumption, and stability through the food and nutrition sub-systems [1]. Disruption of financial exchanges, transportation, and closing of stores led to reduced production, processing, and distribution sub-systems. Rising unemployment, quitting some quarantined jobs, increasing medical healthcare costs, and increasing food basket prices in the consumption sub-system result in lower access to required energy and nutrients, especially in the lower-income groups. Decreased immunity level, increased micronutrient deficiency, overweight, obesity, and non-communicable diseases would also occur [2, 3].

COVID-19 creates an expected "income shock" to increase the prevalence of food-insecure Canadian households. Despite some demand and supply chain disruptions, a broad and rapid appreciation of food prices was not observed. These conditions show the capacity of the Canadian food system to ensure food availability in the short term. Based on the researchers' suggestions, three ongoing strategies, ease of capital flows, international exchange, and maintaining transportation, can help ensure long-term food availability [4]. In the USA, preliminary results of the impact of COVID-19 showed nearly a one-third increase in household food insecurity. Subjects who experienced a job loss were at a higher likelihood of experiencing food insecurity, access challenges, and utilizing coping strategies. Important potential impacts on individual health, including mental health and malnutrition, and future healthcare costs have also been predicted [5].

There is a particular concern for the Western Cape in Africa regarding the short-and long-term shortage of food supply in domestic markets, fertilizers and plant protection products, and food insecurity in vulnerable communities. Monitoring food access in rural areas, especially remote areas, controlling inflation in food prices, direct and indirect assistance to the most vulnerable household can be helpful in the short term. In the long term, the expansion in the production of organic fertilizer on the farm regulates domestic food production chains and coordinates industries, importers/suppliers for the basic goods can improve food security in African households. Despite potentially adverse outcomes of the Covid-19 pandemic, it has highlighted the importance of sustainable food production for the long-term country sustainability [6, 7].

During the early stage of the COVID-19 pandemic period in Iraq, food availability remained stable due to steady international food trade flows and favorable domestic production. Basic food prices did not change significantly; however, vegetable–particularly tomato—prices fluctuated wildly. Importing from different sources, investing in a food security early warning system, and supporting social protection policies can increase the resilience of Iraq's agriculture and food system to current and future shocks. These conditions can also be accounted as an opportunity to introduce digital innovation to improve food security [8]. The

poverty rate increased from 9.4% in 2019 to 9.8% in 2020 after the COVID-19 outbreak, primarily in Indonesian urban areas. Household consumption expenditure decreased by 5.5%, mainly due to the implementation of large-scale social distancing policies in many regions, business closures, lockdown, and movement restrictions. The Government of Indonesia continued supporting the most vulnerable groups through social protection programs. The Ministry of Agriculture has been implementing its subsidized credit scheme program (KUR) to support the agricultural sector [9].

Some studies reported reduced consumption of several food groups, including meat, white roots, and dark leafy greens, in Iranian households during the initial phase of the COVID-19 pandemic. Personal saving rate, occupation status, household income, and nutritional knowledge and skills of household heads were the main socio-economic status (SES) determinants of household food insecurity during the COVID-19 pandemic. Several suggested strategies to improve food security during a global pandemic, such as COVID-19, were e-commerce development, distributing free food baskets to poor households, nutrition education through media, and supporting affected people [10].

## 1.2. Why the review is important

Given the unprecedented nature of the COVID-19 crisis and the importance of early implementation of prevention programs, it is essential to understand better its potential impacts on various dimensions and indicators of food security. The most basic step in making a final judgment about the possible effects of the current crisis on the population's food security is to refer to the available evidence and studies and conduct a systematic review of related studies [11]. So, in this study, the critical food security indicators affected by this crisis will be identified in all populations, especially at-risk populations, to design effective interventions for maintaining and improving the food security status of all people under these conditions. The results of this evidence can provide the information needed to plan food and nutrition security programs and the factors influencing the successful implementation of these programs.

**1.2.1. Objectives.** The current systematic review aims to assess the association between the COVID-19 in its pandemic or endemic phase food security and its indicators, including availability, access, utilization, and stability at individual and household levels in different countries based on WHO-classified regions.

## 2. Methods

## 2.1. Study registration

This review will be conducted using protocol design, search strategy, synthesis, and reporting results from the systematic review guided by the Cochrane Collaboration Handbook of Systematic Reviews [12], the Preferred Reporting Items for Systematic Review, and Meta-Analysis Statement (PRISMA) [13]. The recommended items of completed in the Preferred Reporting Items for Systematic Review and Meta-Analysis Protocols (PRISMA-P) checklist [14] are presented in S2 File. This study was approved by ethics committee of the National Nutrition and Food Technology Research Institute, Shahid Beheshti, University of Medical Sciences (No: IR. SBMU.NNFTRI.REC.1399.041). The present study registration number in the prospective international register of systematic reviews (PROSPERO) is CRD42020185843.

## 2.2. Inclusion criteria for study selection

Study inclusion criteria based on the PICO elements (Population, Intervention, Comparator (s)/Control, and Outcome(s)) are presented in Table 1.

**Table 1. Study inclusion criteria based on PICO elements (18).**

| Inclusion criteria | |
|---|---|
| **Population** | All people of any age, as well as socio-economically disadvantaged groups |
| **Intervention** | COVID-19 is considered an intervention factor |
| **Comparator(s)/ Control** | Not applicable |
| **Outcome(s)** | **Primary outcomes** |
| | a) Food insecurity score and/or prevalence based on validated perception-based measures |
| | b) Food security dimensions and its components: |
| | 1) **Availability** |
| | • Average dietary energy supply adequacy |
| | • The average value of food production |
| | • Dietary energy supply from cereals, roots, and tubers |
| | • Average of protein supply |
| | • The average supply of animal protein supply |
| | 2) **Access** |
| | • Per capita gross domestic product (GDP) in purchasing power equivalent |
| | • The domestic food price index |
| | • Undernourishment prevalence |
| | • The ratio of food expenditure of the poor to total expenditure |
| | • Depth of the food deficit |
| | • Food inadequacy prevalence |
| | 3) **Utilization** |
| | • Wasting percent in under five years children |
| | • Stunting percent in under five years children |
| | • Underweight percent in adults and under five years of children |
| | • Anemia prevalence in pregnant women and under five years of children |
| | • Vitamin A deficiency prevalence |
| | 4) **Stability** |
| | • Cereal import dependency ratio |
| | • Value of food imports over total exports |
| | • Political stability and non-violence/terrorism |
| | • Volatility in domestic food price |
| | • Variability of per capita food production variability |
| | • Variability of per capita food supply variability |
| | **Secondary outcomes:** |
| | • The proportion of anxiety or depression, morbidity, and adverse outcomes, including the proportion of overweight/obese as a potential adverse consequence of the COVID-19 pandemic |
| **Study design** | Community trials and observational studies, including cross-sectional, case-control, and longitudinal studies |
| **Other** | Published in the English language |

Adapted from ref No. [16].

**2.2.1. Type of studies.** The authors will review all studies related to the change in food security status and/or its indicators due to the COVID-19 pandemic and related interventions. Therefore, all field and community trials and observational studies such as cross-sectional, case-control, and longitudinal studies will be reviewed.

**2.2.2. Type of populations.** All population groups, such as children and adults, as well as disadvantaged groups, will be included. The disadvantaged group is a group of people in

vulnerable situations, including low-income people experiencing or at risk of poverty, social exclusion, or discrimination in its multiple dimensions e.g., immigrants and race/ethnic minorities [15].

**2.2.3. Types of interventions.** In the current study, COVID-19 is considered an intervention factor. Food security and/or its indicators at the individual, household, or country level are evaluated as factors influenced by the COVID -19 pandemic. The results will be presented and interpreted at the regional level based on the WHO classification [16].

**2.2.4. Types of outcomes of interests.** Considering the complex nature of food security, we will assess outcomes at different levels, including national, household, and individual. The results of our preliminary search revealed considerable various outcomes across food security and COVID-19. As a result, we will include a structured approach of the outcomes according to the framework of food security definition [16, 17]. Primary outcome measures are considered as the indicators of food security dimensions presented in Table 1.

Secondary outcomes can be proposed as the proportion of anxiety or depression, morbidity, and adverse outcomes, including the proportion of overweight/obese as a potential adverse consequence of the COVID-19 pandemic if they have been reported.

## 2.3. Searching strategies and sources

A sensitive search strategy developed by a combination of words, phrases, and terms related to the possible outcome measures will be used. We work closely with an experienced librarian or information specialist to advise and implement the search strategy. According to the PICO format (Population, Intervention, Comparator(s)/Control, and Outcome(s)) [18] and the MeSH database, a draft of the MEDLINE search strategy for PubMed is presented in S1 File. The following electronic databases will be searched from December 2019 onwards for relevant studies:

- Cochrane Central Register of Controlled Trials (CENTRAL) (The Cochrane Library)

- Cochrane Public Health Register,

- PubMed/MEDLINE,

- SCOPUS

- EMBASE,

- CINAHL,

- Web of science

- Google Scholar

Nearing the completion of our review, we will update the search and include any further eligible studies.

Other relevant studies will be identified by searching in the reference list of included studies, any systematic reviews, hand searching key journals not indexed in the electronic databases, and experts in the field. Reports and unpublished studies will be searched in the grey literature database [19–25].

## 2.4. Study selection

After searching for available scientific sources, articles that report the relationship between the COVID-19 pandemic and the food security of individuals, households, and countries in different groups will be collected. We will use EndNote software to manage the retrieved records

and to remove duplicates. The Systematic Review Assistant-Deduplication Module (SRA-DM) will also be used to validate the de-duplication process [26].

Two people independently review the titles and abstracts of articles using the inclusion criteria checklist. In case of disagreement, the inclusion decision of the article will be finalized through discussion and exchange of views between the research team. At this stage of the screening, irrelevant items will be removed according to the title and abstract. The study selection process will be described through a Preferred Reporting Items for Systematic Reviews and Meta-Analyses (PRISMA) flow chart [13]. Then, two researchers will reread the full text of the articles separately and will decide on including them based on the checklist of inclusion criteria.

## 2.5. Data extraction and management

Two authors (AD and FMN) will extract data independently on a standardized data extraction form. Extracted data will include study characteristics (author (s), publication year and language, study design, setting, and time frame), population characteristics (sample size, age, and sex of subjects), and food insecurity outcomes (change in availability, access, utilization, and stability indicators, due to the COVID-19 pandemic).

## 2.6. Assessment of risk of bias (quality), heterogeneity, and publication bias

In this study, four critical appraisal tools will be used to assess the quality of included studies according to the study design [27]: The Joanna Briggs Institute (JBI) Prevalence Critical Appraisal tool [28], the JBI critical appraisal checklist for randomized control/pseudo-randomized trials, descriptive/case series, and comparable cohort/case-control. These tools were initially designed for use in systematic reviews. Assessment of the data qualitywill be done by one reviewer, and a second reviewer will check it. Any disagreement will be resolved through discussion between the reviewers, and a third reviewer will be consulted if necessary.

Heterogeneity will be tested by the $Chi^2$ statistic [29] and the $I^2$ statistic; $P < 0.1$ for the Chi-square test and an $I^2$ of 75% and above indicates substantial heterogeneity, respectively. Funnel plots were used visually to assess the likelihood of reporting bias of asymmetry sources due to small-study effects, and publication bias for each outcome with ten or more included studies in a meta-analysis. In order to lower the impact of publication bias produced by the remaining studies that cause a funnel plot's asymmetry on the overall effect estimate, the trim-and-fill method will be used [24].

## 2.7. Data analysis

For continuous outcomes in which baseline data are available, we will report the mean difference (MD) between the change in food security and/or its indicators before and after the COVID-19 pandemic if all studies have used the same measures for the outcomes. If the same continuous outcomes have been measured in different ways by different studies, we will use the standardized mean difference (SMD) while reporting 95% confidence intervals (CIs) alongside all effect estimates. If there is considerable heterogeneity ($I^2 > 75\%$), we will only synthesize the results narratively. Meta-analyses in Stata software, version 11 (StataCorp, TX) will be carried out separately for each outcome and type of study design, if the included studies are sufficiently homogeneous ($I^2$ statistic $< 75\%$). We will use the random-effects model for all analyses to incorporate any existing heterogeneity and generate a forest plot for each comparison.

A narrative synthesis of the results will be done by grouping our findings by the type of studies, study population, context, and outcome measurement If possible, the subgroup

analyses will be done for the outcomes between geographic location (e.g., urban/rural, country or region), sex (male/female), age groups (elderly, adults, children, and infants), and socioeconomic status (low, medium, and high SES groups). If it is likely that small-study effects cause asymmetry, we will conduct sensitivity analyses to explore how this affects the meta-analysis results interpretation. In addition, we will explore the impact of blinding, randomization, and studies at high risk of bias on the meta-analyses results.

## 3. Discussion

The initial review of the effects of the COVID-19 pandemic on food security shows that governments have often succeeded in providing enough food supply (availability), but they acted differently in population accessibility to food and its price stability. It seems that differences in the impact of the pandemic on countries' food security are based on the development and stability their food systems; however, this interpretation of data needs to be more studied and examined more closely in different countries with different degrees of development and speeding of the disease.

The COVID-19 pandemic and the consequent lockdowns have considerably influenced people's mental health, particularly those with pre-existing conditions (e.g., eating disorders) [30]. However, some data suggest that the severity of food insecurity is linked to mental health status, particularly in low-income countries [31]. General strategies like supporting the vulnerable groups through social protection programs suggested and used in many countries to prevent increasing food insecurity in the COVID-19 epidemic; while different countries, depending on their circumstances, may need localized and specific solutions. Some unexpected findings from countries [10] can be attributed to the limitations of the studies, especially the generalizability of the sample to the study population, which should be improved in future studies.

In the present conditions, international organizations and developed countries should help low- and middle-income countries to provide the capacity to expand health and social support programs, strengthen food supply chains, and ensure adequate and affordable food sources with the necessary fiscal space and import [32]. While some economic strategies, e.g., social support, can enhance people's ability to cope with vulnerability to food insecurity during COVID-19, international sanctions can make achieving these solutions very difficult or even impossible [33].

The current systematic review results can be used in predicting the effect of COVID-19 on the food security of individuals and households, especially in vulnerable populations, and developing effective interventions. This review can provide policymakers with the information to better understand the factors influencing the implementation of these interventions and inform decision-making to improve food security in other epidemics.

### 3.1. Possible limitations

Numerous articles and reports on food security and COVID-19 pandemics in different countries of the world make it difficult to summarize and draw conclusions from them. For this reason, the project managers decide to categorize the studies based on different regions and provide an analysis in each area to compare them.

## Supporting information

**S1 File. MEDLINE search strategy (via PubMed).**
(DOCX)

**S2 File. PRISMA-P checklist.**
(DOCX)

## Acknowledgments

The authors would like to thank Sepideh Alibeik and Maryam Saryazdi (Academic Support Librarians, National Nutrition and Food Technology Research Institute) for the support with developing the search strategy.

## Author Contributions

**Conceptualization:** Azam Doustmohammadian, Fatemeh Mohammadi-Nasrabadi.

**Data curation:** Azam Doustmohammadian, Fatemeh Mohammadi-Nasrabadi.

**Investigation:** Azam Doustmohammadian, Fatemeh Mohammadi-Nasrabadi.

**Methodology:** Azam Doustmohammadian, Fatemeh Mohammadi-Nasrabadi, Ghasem Fadavi.

**Project administration:** Azam Doustmohammadian, Fatemeh Mohammadi-Nasrabadi.

**Resources:** Fatemeh Mohammadi-Nasrabadi.

**Validation:** Azam Doustmohammadian, Fatemeh Mohammadi-Nasrabadi, Ghasem Fadavi.

**Writing – original draft:** Azam Doustmohammadian, Fatemeh Mohammadi-Nasrabadi.

**Writing – review & editing:** Azam Doustmohammadian, Fatemeh Mohammadi-Nasrabadi, Ghasem Fadavi.

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
