## [Decision Letter · Decision Letter 0]

16 May 2022

PONE-D-21-32703COVID-19 pandemic and food security in different contexts: a systematic review protocolPLOS ONE

Dear Dr. Mohammadi-Nasabadi  ,

Thank you for submitting your manuscript to PLOS ONE. After careful consideration, we feel that it has merit but does not fully meet PLOS ONE’s publication criteria as it currently stands. Therefore, we invite you to submit a revised version of the manuscript that addresses the points raised during the review process.

We look forward to receiving your revised manuscript.

Kind regards,

Negar Rezaei, M.D., Ph.D.,

Academic Editor

PLOS ONE

**Journal requirements:**

“This systematic review is funded by National Nutrition and Food Technology Research Institute, Shahid Beheshti University of Medical Sciences (Grant No. 99-23953).”

“The funders had and will not have a role in study design, data collection and analysis, decision to publish, or preparation of the manuscript.”

**Reviewers' comments:**

Reviewer's Responses to Questions

**Comments to the Author**

1. Does the manuscript provide a valid rationale for the proposed study, with clearly identified and justified research questions?

Reviewer #1: Yes

Reviewer #2: Partly

Reviewer #3: Partly

Reviewer #4: Yes

2. Is the protocol technically sound and planned in a manner that will lead to a meaningful outcome and allow testing the stated hypotheses?

Reviewer #1: Yes

Reviewer #2: Partly

Reviewer #3: Partly

Reviewer #4: Yes

3. Is the methodology feasible and described in sufficient detail to allow the work to be replicable?

Reviewer #1: Yes

Reviewer #2: No

Reviewer #3: Yes

Reviewer #4: Yes

4. Have the authors described where all data underlying the findings will be made available when the study is complete?

Reviewer #1: Yes

Reviewer #2: No

Reviewer #3: Yes

Reviewer #4: Yes

5. Is the manuscript presented in an intelligible fashion and written in standard English?

Reviewer #1: Yes

Reviewer #2: No

Reviewer #3: No

Reviewer #4: Yes

6. Review Comments to the Author

You may also provide optional suggestions and comments to authors that they might find helpful in planning their study.

Reviewer #1: Doustmohammadian et al. proposed a protocol for a systematic review on the impact of COVID-19 pandemic on food security in different contexts. They described the importance of this review, thoroughly. Also, their search strategy seems reasonable for me. Although I have some comments to address.

1. Section 2.2.1: Please state what are the inclusion criteria.

2. In the discussion section, please elaborate on specific conditions such as eating disorders. Please refer to the meta-analysis by Haghshomar et al. to discuss this part. (https://jeatdisord.biomedcentral.com/articles/10.1186/s40337-022-00550-9)

3. Please reconsider using RevMan, as it is not the most professional software for conducting meta-analysis.

4. Please consider utilizing JBI Critical Appraisal Checklist instead of NOS, to cover all kind of the included studies.

Reviewer #2: Recommendation

Major Revision

Comments to Author

Ms. Ref. No.: PONE-D-21-32703

Title: COVID-19 pandemic and food security in different contexts: a systematic review protocol

Overview

The article “COVID-19 pandemic and food security in different contexts: a systematic review protocol” is a systematic review protocol article. The objective of this protocol is to study the covid impact on different aspects of food insecurity, including availability, accessibility, consumption, and stability. Overall, the article needs to be more specific, needs search syntax revision and needs writing improvement.

Comments

• The outcomes of the study are not clear enough. It is not clear how each indicator of food security is defined. If this is defined based on the outcomes provided in table 1, why authors didn’t use those outcomes in their search syntax? For example, one of the outcomes of the availability is “Adequacy of protein supply”; however, there are no keywords about it in the search syntax.

• In addition, the outcomes presented in table 1 are different from the reference they cited. For example, in the FAO report, the “average protein supply” was mentioned as an indicator of food security. However, here, the “Adequacy of protein supply” is mentioned, which has a different meaning.

• The search syntax needs significant improvement.

1. Using COVID in the filter of search is not comparable to using covid-related words in the search syntax. Your search will be inaccurate if you don’t use COVID in your search syntax. This is because filtering papers to COVID doesn’t necessarily mean the retrieved papers are related to the impact of covid on food security. In addition, filters are based on MeSH terms, and it takes time for each paper to get MeSH. Thus, the recently published papers will not be shown up in your search results.

2. It is not clear how the authors selected the keywords.

3. Parenthesis and quotation marks are needed for many of the terms. For example, instead of searching grocery store[tiab], it is better to search “grocery store”[tiab] or (grocery[tiab] AND store*[tiab]).

• The population of the study should be more specific. The disadvantaged group needs to be identified clearly.

With the current format, this protocol paper is not suitable for publication.

Reviewer #3: COVID-19 pandemic and food security in different contexts: a systematic review protocol

The authors of this study protocol developed a search strategy to assess the impact of COVID-19 pandemic on food security. The conceptualization of the systematic review is robust and the results would benefit policy makers on the prevention food insecurity during emerging diseases. Although the manuscript is designed and drafted well, some comments need to be considered prior to the decision on this submission.

Title:

1. From the context of the manuscript this study includes meta-analysis as well, please consider mentioning it in the title if this is correct.

Introduction:

2. Although the effect of food insecurity caused by pandemic could manifest itself with delay, but we have entered the endemic phase of COVID-19. Therefore, it is suggested that the authors revise the aim of the study accordingly. The results could be of value for policy making for other epidemics.

3. Line 110-111. The target population for this study is “At-risk population”. However, in lines 135-137 of the method the population of the study is “all groups”. Please revise line 111.

Method:

4. Line 136: Please clarify the scope of the study here. Throughout the text different words such as individual, household, country, regional and global are mentioned. Also, If the study is on the regional level please mention what is categorization source (e.g. World Bank, WHO, …)

5. Line 136: It is suggested that “disadvantaged groups” to be defined here and have their own section in the search strategy table as well.

6. Line 174: Some of the reference website provided for grey literature are not accessible (e.g. https://www.nyam.org/library/online-resources/grey-literature-report/). They could also be presented at the Reference section and not at the main body of the manuscript.

Discussion:

7. Assuming there are some limitations to the study, the authors could mention them

Search strategy

8. The search terms in the table 2 of the appendix seems to be chosen carefully. However, based on the indicators that were presented at Table 1 there are four categories of the food securities that lack of each has several outcomes. It seems that the results for some outcomes could not be extracted from the search results with the current search strategy (e.g. the utilization category; wasting, stunting, anemia and underweight outcomes)

Minor comments:

1. There were some typos across the manuscript (e.g. line: 211, 215, 223, or in the appendix, search strategy table, No.8, word oradequate)

Appendix:

1. The table of search terms is the first one in the appendix, please renumber the table to Table 1.

Reviewer #4: 1. in the method section, In order to detect and adjust for publication bias, the trim-and-fill method can be used. To lower the impact of publication bias produced by the remaining studies that cause a funnel plot's asymmetry on overall effect estimate.

2. In the introduction section, line 59-60, ""Increased micronutrient deficiency and decreased immunity level, increased overweight, obesity, and non-communicable diseases would also occur "" it may more apprehensible to be written in stratified format, reporting increased items and then decreased one.

7. PLOS authors have the option to publish the peer review history of their article (what does this mean?). If published, this will include your full peer review and any attached files.

Reviewer #1: No

Reviewer #2: No

Reviewer #3: No

Reviewer #4: **Yes: **Seyedeh Melika Hashemi

---

## [Author Response · Author response to Decision Letter 0]

21 Jun 2022

Dear editor and reviewers

Greetings

We would like to thank you for your thoughtful comments. Accordingly, the authors have revised the manuscript point-by-point based on the below comments, checked our manuscript to meet PLOS ONE's style requirements, provided the correct grant numbers for the awards we received for our study in the ‘Funding Information’ section, removed any funding-related text from the manuscript and updated our “Funding Statemen”. We moved ethics statement to Methods section of our manuscript, too and uploaded two files for revision including the track changed, and the final version of the cleaned manuscript. The page and line numbers correspond to the highlighted file of the revised manuscript. The manuscript was totally edited and paraphrased as well. We hope that the editor feels we have adequately addressed all of the comments, and look forward to hearing the outcome in due course. 

Best regards

Dr. Fatemeh Mohammadi-Nasrabadi

Corresponding author

Reviewer #1: 

Doustmohammadian et al. proposed a protocol for a systematic review on the impact of COVID-19 pandemic on food security in different contexts. They described the importance of this review, thoroughly. Also, their search strategy seems reasonable for me. Although I have some comments to address.

1. Section 2.2.1: Please state what are the inclusion criteria.

Authors' Answer: Study inclusion criteria based on the PICO elements (Population, Intervention, Comparator(s)/Control, and Outcome(s)) are presented in Table 1.

Previous Table 1 was also merged in the following Table 1 (page 8, Table 1).

Corrections were track changed on page 5-7, line 133-162.

Table 1: Study inclusion criteria based on PICO elements (18)

Inclusion criteria

Population All people of any age, as well as socio-economically disadvantaged groups

Intervention COVID-19 is considered an intervention factor

Comparator(s)/Control Not applicable

Outcome(s) Primary outcomes

a) Food insecurity score and/or prevalence based on validated perception-based measures 

b) Food security dimensions and its components:

1) Availability

• Average dietary energy supply adequacy 

• Average value of food production 

• Dietary energy supply from cereals, roots and tubers

• Average of protein supply

• Average supply of animal protein supply

2) Access

• Per capita gross domestic product (GDP) in purchasing power equivalent

• Domestic food price index

• Undernourishment prevalence 

• Ratio of food expenditure of the poor to total expenditure

• Depth of the food deficit

• Food inadequacy prevalence

3) Utilization

• Wasting percent in under 5 years children

• Stunting percent in under 5 years children

• Underweight percent in adults and under 5 years children 

• Anemia prevalence in pregnant women and under 5 years children 

• Vitamin A deficiency prevalence

4) Stability

• Cereal import dependency ratio

• Value of food imports over total exports

• Political stability and non- violence/terrorism

• Volatility in domestic food price 

• Variability of per capita food production variability

• Variability of per capita food supply variability

Secondary outcomes: 

• Proportion of anxiety or depression, morbidity, and adverse outcomes, including the proportion of overweight/obese as a potentially adverse consequence of the COVID-19 pandemic

Study design Community trials and observational studies, including cross-sectional, case-control and longitudinal studies

Other Published in English language

Adapted from ref No.(16)

2. In the discussion section, please elaborate on specific conditions such as eating disorders. Please refer to the meta-analysis by Haghshomar et al. to discuss this part. (https://jeatdisord.biomedcentral.com/articles/10.1186/s40337-022-00550-9)

Authors' Answer: It was corrected as follows on pages 10-11, lines 256-260.

“The COVID-19 pandemic and the consequent lockdowns have considerably influenced people's mental health, particularly those with pre-existing conditions (e.g., eating disorders) (30). However, some data suggest that the severity of food insecurity is linked to mental health status, particularly in low-income countries (31)."

3. Please reconsider using RevMan, as it is not the most professional software for conducting meta-analysis.

Authors' Answer: Thanks for the reviewer's comment. It was corrected as "Stata software, version 11 (StataCorp, TX)" on page 10, lines 235-236.

4. Please consider utilizing JBI Critical Appraisal Checklist instead of NOS, to cover all kind of the included studies.

Authors' Answer: Thanks for the reviewer's constructive comment. It was corrected accordingly on page 9, lines 215-222. 

"In this study, four critical appraisal tools will be used to assess the quality of included studies according to the study design (27). The Joanna Briggs Institute (JBI) Prevalence Critical Appraisal tool (28), the JBI critical appraisal checklist for randomized control/pseudo-randomized trials, descriptive/case series, and comparable cohort/case-control. These tools were designed initially for use in systematic reviews (29)."

Reviewer #2: Recommendation

Major Revision

Comments to Author

Ms. Ref. No.: PONE-D-21-32703

Title: COVID-19 pandemic and food security in different contexts: a systematic review protocol

Overview

The article “COVID-19 pandemic and food security in different contexts: a systematic review protocol” is a systematic review protocol article. The objective of this protocol is to study the covid impact on different aspects of food insecurity, including availability, accessibility, consumption, and stability. Overall, the article needs to be more specific, needs search syntax revision and needs writing improvement.

Comments

1) The outcomes of the study are not clear enough. It is not clear how each indicator of food security is defined. If this is defined based on the outcomes provided in table 1, why authors didn’t use those outcomes in their search syntax? For example, one of the outcomes of the availability is “Adequacy of protein supply”; however, there are no keywords about it in the search syntax.

Authors' Answer: The search strategy will be further developed during the study, and the previous search strategy was replaced with the new one in Appendix 1.

2) In addition, the outcomes presented in table 1 are different from the reference they cited. For example, in the FAO report, the “average protein supply” was mentioned as an indicator of food security. However, here, the “Adequacy of protein supply” is mentioned, which has a different meaning.

Authors' Answer: Thanks for the reviewer's comment. The outcomes presented in Table 1 were matched with the reference cited (FAO report). It was corrected accordingly on Table 1, page 6-7, lines 160-161. 

3) The search syntax needs significant improvement.

3.1. Using COVID in the filter of search is not comparable to using covid-related words in the search syntax. Your search will be inaccurate if you don’t use COVID in your search syntax. This is because filtering papers to COVID doesn’t necessarily mean the retrieved papers are related to the impact of covid on food security. In addition, filters are based on MeSH terms, and it takes time for each paper to get MeSH. Thus, the recently published papers will not be shown up in your search results.

3.2. It is not clear how the authors selected the keywords.

3.3 Parenthesis and quotation marks are needed for many of the terms. For example, instead of searching grocery store[tiab], it is better to search “grocery store”[tiab] or (grocery[tiab] AND store*[tiab]).

Authors' Answer: According to the PICO format (Population, Intervention, Comparator(s)/Control, and Outcome(s)) (18) and the MeSH database, a draft of the MEDLINE search strategy for PubMed is presented in Appendix 1. (please see page 7, lines 167-170 & Appendix 1) 

The search strategy will be further developed during the study and the previous search strategy was replaced with the new one in Appendix 1.

4) The population of the study should be more specific. The disadvantaged group needs to be identified clearly.

Authors' Answer: Disadvantaged group was defined as follows (page 5, line 142-144). It was added to search strategy, as well:

“Disadvantaged group is a group of people in vulnerable situations, including low income people experiencing or at risk of poverty, social exclusion or discrimination in its multiple dimensions e.g. immigrants and race/ethnic minorities.”

5) With the current format, this protocol paper is not suitable for publication.

Authors' Answer: The authors have revised the manuscript point-by-point based on the reviewers' comments. We hope that the dear editor and reviewers feel we have adequately addressed all of the comments.

 

Reviewer #3: COVID-19 pandemic and food security in different contexts: a systematic review protocol

The authors of this study protocol developed a search strategy to assess the impact of COVID-19 pandemic on food security. The conceptualization of the systematic review is robust and the results would benefit policy makers on the prevention food insecurity during emerging diseases. Although the manuscript is designed and drafted well, some comments need to be considered prior to the decision on this submission.

Title:

1. From the context of the manuscript this study includes meta-analysis as well, please consider mentioning it in the title if this is correct.

Authors' Answer: As mentioned in data analysis (page 10, lines 240-242): “Meta-analyses in Stata software, version 11 (StataCorp, TX) will be carried out separately for each outcome and type of study design, if the included studies are sufficiently homogeneous (I2 statistic<75%)”; so, it was not included in the title due to uncertainty about conducting Meta-analysis.

Introduction:

2. Although the effect of food insecurity caused by pandemic could manifest itself with delay, but we have entered the endemic phase of COVID-19. Therefore, it is suggested that the authors revise the aim of the study accordingly. The results could be of value for policy making for other epidemics.

Authors' Answer: The objectives were changed as follows based on the reviewer’s comment (page 5, lines 118-121):

“The current systematic review aims to assess the association between the COVID-19 in its pandemic or endemic phase, food security and its indicators, including availability, access, utilization, and stability at individual and household levels in different countries based on WHO classified regions.”

3. Line 110-111. The target population for this study is “At-risk population”. However, in lines 135-137 of the method the population of the study is “all groups”. Please revise line 111.

Authors' Answer: The mentioned sentence was rewritten as follows (page 4, line 113): 

“So, in this study, the critical food security indicators affected by this crisis will be identified in all population especially at-risk populations, to design effective interventions toward maintaining and improving the food security status of all people under these conditions.”

Method:

4. Line 136: Please clarify the scope of the study here. Throughout the text different words such as individual, household, country, regional and global are mentioned. Also, If the study is on the regional level please mention what is categorization source (e.g. World Bank, WHO, …)

Authors' Answer: The following explanation was added (page 6, lines 153-154) based on reviewer’s comment:

“The results will be presented and interpreted at regional level based on the WHO classification.”

5. Line 136: It is suggested that “disadvantaged groups” to be defined here and have their own section in the search strategy table as well.

Authors' Answer: Disadvantaged group was defined as follows (page 5, line 142-144), and added to search strategy:

“Disadvantaged group is a group of people in vulnerable situations, including low income people experiencing or at risk of poverty, social exclusion or discrimination in its multiple dimensions e.g. immigrants and race/ethnic minorities.”

6. Line 174: Some of the reference website provided for grey literature are not accessible (e.g. https://www.nyam.org/library/online-resources/grey-literature-report/). They could also be presented at the Reference section and not at the main body of the manuscript.

Authors' Answer: The reference website was corrected as follows and transferred to Reference section based on the reviewer’s comment:

“https://www.nyam.org/library/collections-and-resources/grey-literature-report/”

Discussion:

7. Assuming there are some limitations to the study, the authors could mention them

Authors' Answer: The possible limitations were added to the end of the Discussion section based on the reviewer’s comment (page 11, lines 283-287):

“Numerous articles and reports on food security and COVID-19 pandemics in different countries of the world make it difficult to summarize and draw conclusions from them. For this reason, the project managers decide to categorize the studies based on different regions and provide an analysis in each area to compare them.”

Search strategy

8. The search terms in the table 2 of the appendix seems to be chosen carefully. However, based on the indicators that were presented at Table 1 there are four categories of the food securities that lack of each has several outcomes. It seems that the results for some outcomes could not be extracted from the search results with the current search strategy (e.g. the utilization category; wasting, stunting, anemia and underweight outcomes).

Authors' Answer: The search strategy will be further developed during the study and the previous search strategy was replaced with the new one in Appendix 1. 

Minor comments:

1. There were some typos across the manuscript (e.g. line: 211, 215, 223, or in the appendix, search strategy table, No.8, word oradequate)

Authors' Answer: All of the typos in the manuscript were corrected based on the reviewer’s comment.

Appendix:

1. The table of search terms is the first one in the appendix, please renumber the table to Table 1.

Authors' Answer: In the revised manuscript Table , Study inclusion criteria is the first one in the manuscript and the table of search terms was numbered as Table 2 in Appendix.

Reviewer #4: 

1. in the method section, In order to detect and adjust for publication bias, the trim-and-fill method can be used. To lower the impact of publication bias produced by the remaining studies that cause a funnel plot's asymmetry on overall effect estimate.

Authors' Answer: The following sentence was added to method section based on the reviewer’s comment (page 10, lines 230 -232):

“In order to lower the impact of publication bias produced by the remaining studies that cause a funnel plot's asymmetry on overall effect estimate, the trim-and-fill method will be used.”

2. In the introduction section, line 59-60, ""Increased micronutrient deficiency and decreased immunity level, increased overweight, obesity, and non-communicable diseases would also occur "" it may more apprehensible to be written in stratified format, reporting increased items and then decreased one.

Authors' Answer: The mentioned sentence was modifies based on the reviewer’s comment as follows (pagew 3, lines 60-61):

“Decreased immunity level, increased micronutrient deficiency, overweight, obesity, and non-communicable diseases would also occur.”

---

## [Decision Letter · Decision Letter 1]

2 Aug 2022

COVID-19 pandemic and food security in different contexts: a systematic review protocol

PONE-D-21-32703R1

Dear Dr. Mohammadi-Nasrabadi,

We’re pleased to inform you that your manuscript has been judged scientifically suitable for publication and will be formally accepted for publication once it meets all outstanding technical requirements.

Kind regards,

Negar Rezaei, M.D., Ph.D.,

Academic Editor

PLOS ONE

Additional Editor Comments (optional):

Reviewers' comments:

Reviewer's Responses to Questions

**Comments to the Author**

1. Does the manuscript provide a valid rationale for the proposed study, with clearly identified and justified research questions?

Reviewer #1: Yes

Reviewer #3: Yes

2. Is the protocol technically sound and planned in a manner that will lead to a meaningful outcome and allow testing the stated hypotheses?

Reviewer #1: Yes

Reviewer #3: Yes

3. Is the methodology feasible and described in sufficient detail to allow the work to be replicable?

Reviewer #1: Yes

Reviewer #3: Yes

4. Have the authors described where all data underlying the findings will be made available when the study is complete?

Reviewer #1: Yes

Reviewer #3: Yes

5. Is the manuscript presented in an intelligible fashion and written in standard English?

Reviewer #1: Yes

Reviewer #3: Yes

6. Review Comments to the Author

You may also provide optional suggestions and comments to authors that they might find helpful in planning their study.

Reviewer #1: Thanks. The revised version is now suitable for publication. There are no more recommendations from me.

Reviewer #3: The results of this study will be of interest for public health authorities working on food security area. The revised manuscript has addressed all comments and no further revision is needed.

7. PLOS authors have the option to publish the peer review history of their article (what does this mean?). If published, this will include your full peer review and any attached files.

Reviewer #1: No

Reviewer #3: **Yes: **Rosa Haghshenas

---

## [Editor Report · Acceptance letter]

2 Sep 2022

PONE-D-21-32703R1 

COVID-19 pandemic and food security in different contexts: a systematic review protocol 

Dear Dr. Mohammadi-Nasrabadi:

I'm pleased to inform you that your manuscript has been deemed suitable for publication in PLOS ONE. Congratulations! Your manuscript is now with our production department. 

Kind regards, 

on behalf of

Dr. Negar Rezaei 

Academic Editor

PLOS ONE